# FEDRACE: A Hierarchical and Statistical Framework for Robust Federated Learning

**Gang Yan**    **Sikai Yang**    **Wan Du**
University of California, Merced
Merced, CA, United States
{gyan5, syang126, wdu3}@ucmerced.edu

## Abstract

Integrating large pre-trained models into federated learning (FL) can significantly improve generalization and convergence efficiency. A widely adopted strategy freezes the pre-trained backbone and fine-tunes a lightweight task head, thereby reducing computational and communication costs. However, this partial fine-tuning paradigm introduces new security risks, making the system vulnerable to poisoned updates and backdoor attacks. To address these challenges, we propose FEDRACE, a unified framework for robust FL with partially frozen models. FEDRACE comprises two core components: **HStat-Net**, a hierarchical network that refines frozen features into compact, linearly separable representations; and **DevGuard**, a server-side mechanism that detects malicious clients by evaluating statistical deviance in class-level predictions modeling generalized linear models (GLMs). DevGuard further incorporates adaptive thresholding based on theoretical misclassification bounds and employs randomized majority voting to enhance detection reliability. We implement FEDRACE on the FedScale platform and evaluate it on CIFAR-100, Food-101, and Tiny ImageNet under diverse attack scenarios. FEDRACE achieves a true positive rate of up to 99.3% with a false positive rate below 1.2%, while preserving model accuracy and improving generalization.

## 1 Introduction

Federated learning (FL) [1, 2, 3] enables multiple clients to collaboratively train a global model while keeping data on-device. By keeping raw data local, this decentralized approach preserves privacy, making it well-suited for privacy-sensitive applications such as voice recognition, healthcare, and human activity monitoring [4, 5, 6]. Recent advances have shown that integrating large pre-trained models into FL can significantly improve generalization and convergence speed. Models such as CLIP [7] and BERT [8] act as powerful feature extractors across various domains and tasks. Their ability to encode rich, transferable knowledge is particularly valuable in federated settings, where data distributions across clients are typically non-IID (Independent and Identically Distributed) [9, 10].

While pre-trained models enhance generalization in federated learning, fine-tuning the entire model on each client is often impractical due to limited computational and communication resources. To address this, a widely adopted approach is to freeze the pre-trained backbone and fine-tune only a small, task-specific head [8, 11, 12, 13]. This strategy maintains the utility of large models while reducing training overhead. However, this partial adaptation approach introduces new security risks. Since the backbone is fixed and shared across clients, adversaries can exploit its stable representation space to introduce poisoned updates or embed backdoors [14, 15, 16]. Our experiments reveal that untargeted attacks [17] can reduce model accuracy by over 11.7%, while distributed backdoor attacks [18] can achieve a success rate exceeding 80% with minimal impact on clean model performance.

39th Conference on Neural Information Processing Systems (NeurIPS 2025).

Existing defenses such as Trimmed-Mean [19], Multi-Krum [20], and reputation-based methods like FLShield [21] and FLAIR [22] often rely on gradient statistics or fixed heuristics. While these methods are effective in some settings, they struggle to detect subtle semantic manipulations, especially when only the head is trainable [23, 24]. These limitations raise a key research question:

*How can we integrate large pre-trained models into FL while enabling reliable and adaptive detection of malicious clients?*

To answer this question, we propose FEDRACE, a unified framework for **Fed**erated **R**epresentation-based **A**daptive **C**lient **E**valuation. FEDRACE combines hierarchical representation learning with statistical client evaluation to improve FL robustness. It consists of two main components: (1) **HStat-Net**, a Hierarchical Statistical Network that transforms fixed features into compact and linearly separable representations using a triplet loss, and (2) **DevGuard**, a server-side evaluation mechanism that uses a generalized linear model (GLM) to identify clients with abnormal semantic behavior through deviance analysis.

As illustrated in Figure 1, HStat-Net is composed of three parts: a frozen pre-trained feature extractor $\phi$, a trainable statistical projection layer **s**, and a lightweight task head **h** implemented as a GLM. This modular design supports efficient fine-tuning and enables interpretable predictions. DevGuard evaluates clients by comparing their prediction outputs to global class-wise embeddings. It calculates log-likelihood deviations to assign deviance scores, providing a principled and explainable measure of client reliability. To improve robustness, DevGuard uses adaptive thresh-

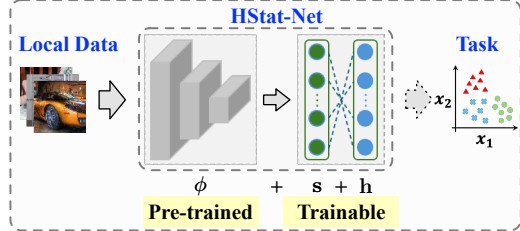

Figure 1: HStat-Net architecture ($\mathbf{w} = \mathbf{h} \circ \mathbf{s} \circ \phi$), supporting robust federated learning. *Local Data* samples are from Tiny ImageNet [25].

olding informed by theoretical misclassification bounds and applies majority voting over client subsets to reduce the impact of noise and adversarial updates.

Our contributions are summarized as follows. First, we propose **HStat-Net**, a hierarchical architecture that enables secure and efficient federated learning by transforming frozen representations into features suitable for statistical detection and model training. Second, we design **DevGuard**, a deviance-based client evaluation mechanism that leverages generalized linear modeling to identify semantic inconsistencies and malicious behaviors in a principled and interpretable manner. Finally, we develop **FEDRACE**, an integrated framework that combines representation refinement with adaptive client evaluation, offering strong security and generalization.

We also implement FEDRACE on the FedScale platform [26] and conduct extensive evaluations on CIFAR-100, Food-101 [27], and Tiny ImageNet [25]. The experiments cover various attack types, including untargeted, targeted, and backdoor scenarios, under realistic conditions. Results show that FEDRACE achieves a true positive rate of up to 99.3% and a false positive rate below 1.2%, while maintaining high model accuracy and fast convergence.

## 2 Background and Motivation

### 2.1 Federated Learning and Pre-trained Models

Federated learning is a distributed training framework that enables multiple clients, denoted by $\mathcal{N} = \{1, \ldots, N\}$, to collaboratively train a global model without sharing raw data. The overall objective is to minimize the global loss:

$$\min_{\mathbf{w}} \mathcal{L}(\mathbf{w}, \mathcal{D}) = \sum_{i=1}^{N} \frac{|\mathcal{D}_i|}{|\mathcal{D}|} \mathcal{L}_i(\mathbf{w}, \mathcal{D}_i), \tag{1}$$

where $\mathcal{D}_i$ is the local dataset of client $i$, and **w** denotes the global model parameters. A commonly used optimization algorithm is FedAvg [1], which proceeds in communication rounds. In each round, the server selects a subset of clients $\mathcal{N}^{(t)}$, distributes the current global model, and aggregates the returned updates through weighted averaging.

Recent work has shown that integrating large pre-trained models into FL can improve generalization and accelerate convergence. Models such as CLIP [7], which builds on the ViT-B/32 backbone [11], provide strong and transferable representations. These models are particularly effective in federated settings, even under non-IID data distributions [9, 10].

A typical deployment strategy is to freeze the pre-trained feature extractor $\phi$ and train only a lightweight, task-specific head $\mathbf{h}$, forming a modular model $\mathbf{w} = \mathbf{h} \circ \phi$. We evaluate this strategy on the CIFAR-100 dataset [28] by

Table 1: Comparison of training strategies on CIFAR-100.

| Method | Trainable Parameters | Training Time | Testing Accuracy |
|---|---|---|---|
| Retrain | $\approx 86.62$M | 0.0423 sec | 58.32% |
| Fully Fine-Tuned | $\approx 86.62$M | 0.0411 sec | 68.04% |
| Partially Fine-Tuned | $\approx 0.05$M | 0.0130 sec | 75.99% |

comparing three training modes: (1) training from scratch, (2) full fine-tuning of all parameters, and (3) partial fine-tuning of only the task head $\mathbf{h}$. As shown in Table 1, partial fine-tuning achieves the highest test accuracy (75.99%) and reduces training time by 68.37% compared to full fine-tuning. These results confirm that updating only the task head is both efficient and effective in FL settings. The effectiveness of partial fine-tuning is largely due to the stable and transferable representations provided by $\phi$. By keeping $\phi$ fixed, we reduce the risk of overfitting and lower the computational and communication overhead, which is critical for practical FL deployments.

## 2.2 Threat Models and Poisoning Strategies

Although federated learning keeps data local, it remains vulnerable to adversarial threats. Malicious clients can manipulate local computations to degrade global performance or induce specific misbehavior. We consider three common attack types:

**Untargeted attacks** aim to reduce overall model accuracy without targeting specific inputs [17, 23, 29, 30]. Their impact is measured by classification accuracy (ACC):

$$\text{ACC} = \mathbb{E}_{(\mathbf{x},y)\sim\mathcal{D}} \left[ \mathbb{I}\left(\mathbf{G}(\mathbf{x}) = y\right) \right], \tag{2}$$

where $y$ is the true label, $\mathbf{G}(\mathbf{x})$ is the model's prediction, and $\mathbb{I}(\cdot)$ indicates correctness.

**Targeted attacks** aim to misclassify selected inputs into attacker-specified labels while maintaining clean accuracy [31, 32, 33, 34]. Their success is measured by the Attack Success Rate (ASR):

$$\text{ASR} = \mathbb{E}_{(\mathbf{x},y)\sim\mathcal{D}_{\text{attack}}} \left[ \mathbb{I}\left(\mathbf{G}(\mathbf{x}) = y^{\text{target}}\right) \right]. \tag{3}$$

**Backdoor attacks** inject triggers into inputs to control model outputs, while keeping performance on clean data unaffected [18, 35, 36, 37, 38]. Effectiveness is measured by Backdoor Accuracy (BA):

$$\text{BA} = \mathbb{E}_{(\mathbf{x},y)\sim\mathcal{D}_{\text{attack}}} \left[ \mathbb{I}\left(\mathbf{G}(\mathbf{T}_{\text{trigger}}(\mathbf{x})) = y^{\text{target}}\right) \right], \tag{4}$$

where $\mathbf{T}_{\text{trigger}}(\cdot)$ applies the backdoor trigger to the input $\mathbf{x}$.

In this work, we adopt a practical threat model where up to $M < 0.5N$ clients may be compromised. This constraint aligns with assumptions in Byzantine-robust FL [23, 24, 30]. Compromised clients may modify local data, manipulate gradients, or submit crafted updates [29, 31, 35, 39, 40]. Unlike stronger threat models [29, 30, 40, 41], we assume adversaries act independently and do not have access to global aggregation logic or benign clients' data.

## 2.3 Existing Defenses in Federated Learning

To address adversarial threats in federated learning, a range of defenses has been proposed to address adversarial threats in federated learning. Early methods such as Trimmed-Mean [19, 40] and Multi-Krum [20] filter anomalous updates based on statistical or geometric criteria. More recent approaches incorporate dynamic analysis or auxiliary signals. For example, FedRoLA [42] measures layer-wise similarity to detect outliers, while FLShield [21] and FLAIR [22] leverage validation feedback and client reputation. Additional methods like WPCRA [43], FedGT [44], and MAB-RFL [45] apply cross-round analysis, group testing, or adaptive client selection.

However, most existing defenses are designed for full-model training and do not address the challenges of partial fine-tuning. In many practical deployments, the feature extractor $\phi$ is frozen and only the task-specific head $\mathbf{h}$ is trained to reduce communication and computation overhead [15]. While

efficient, this setup exposes new vulnerabilities. The shared frozen representation space can be exploited to embed triggers or introduce structured deviations. As shown in [14], even a few poisoned samples can achieve nearly 100% backdoor success with minimal impact on clean accuracy.

Our experiments confirm this risk. On CIFAR-100, an untargeted attack [17] reduces accuracy from 75.99% to 64.24%, while a distributed backdoor attack [18] reaches over 80% backdoor accuracy without significantly affecting clean performance. These findings reveal a key limitation in current FL defenses, which lack effective protection for systems using partial fine-tuning.

## 3 Proposed Design

To mitigate the security vulnerabilities introduced by partial fine-tuning in federated learning, we propose FEDRACE, a unified framework that integrates robust representation learning with adaptive client evaluation. FEDRACE comprises two core components:

1. **HStat-Net**, a hierarchical statistical network that refines frozen features into compact, linearly separable representations adapted to the downstream task.
2. **DevGuard**, a server-side detection mechanism that evaluates client updates based on semantic consistency and statistical deviance under a generalized linear model.

By combining local representation refinement with global statistical evaluation, FEDRACE enhances robustness in non-IID and adversarial federated settings. We now detail each component.

### 3.1 HStat-Net: Hierarchical Statistical Network

To mitigate the limitations of frozen representations in partially fine-tuned FL systems, HStat-Net refines features to improve task adaptation and enhance representation structure. As shown in Figure 1, it adopts a modular architecture composed of three components:

1. **Pre-trained Feature Extractor ($\phi$):** The extractor $\phi$ maps raw input $\mathbf{x}$ to a feature vector $\mathbf{z}$:

$$\mathbf{z} = \phi(\mathbf{x}). \tag{5}$$

   This module is fixed during training to ensure consistency across clients and reduce computation and communication overhead.

2. **Statistical Net ($\mathbf{s}$):** The net transforms $\mathbf{z}$ into a compact, linearly separable representation:

$$\mathbf{r} = \mathbf{s}(\mathbf{z}), \tag{6}$$

   where $\mathbf{z} \in \mathbb{R}^D$ is projected to $\mathbf{r} \in \mathbb{R}^d$ with $d < D$. This transformation reduces dimensionality and abstracts feature information, which can help improve privacy [46, 47].

3. **Task Net ($\mathbf{h}$):** The Task Net produces task-specific predictions from $\mathbf{r}$:

$$\hat{y} = \mathbf{h}(\mathbf{r}). \tag{7}$$

   This component is lightweight and trained locally to support downstream tasks with minimal computational cost.

For each client $i$, the complete forward pipeline is defined as:

$$\hat{y}_i = \mathbf{h}_i(\mathbf{s}_i(\phi(\mathbf{x}))) = \psi_i(\phi(\mathbf{x})), \tag{8}$$

where $\psi_i$ denotes the client-specific transformation applied after the shared extractor $\phi$.

**Training Methodology.** To effectively train the modular components of HStat-Net while preserving their distinct roles, we adopt a two-stage training procedure that decouples the optimization of the Statistical Net ($\mathbf{s}$) and the Task Net ($\mathbf{h}$). To promote a discriminative feature space, the Statistical Net is optimized using the *Triplet Loss* [48, 49], which encourages intra-class compactness and inter-class separation. The loss is defined as:

$$\mathcal{L}_{\text{Triplet}} = \sum_{l \in \mathcal{D}^{\text{batch}}} \max\left( \|\mathbf{r}_l - \mathbf{r}_l^p\|_2^2 - \|\mathbf{r}_l - \mathbf{r}_l^n\|_2^2 + \delta, 0 \right), \tag{9}$$

where $\mathbf{r}_l$ is the representation of anchor $\mathbf{x}_l$, $\mathbf{r}_l^p$ is a positive sample from the same class, $\mathbf{r}_l^n$ is a negative sample from a different class, and $\delta$ is a margin parameter, empirically set to 0.1.

The Task Net is trained to optimize predictive performance using the standard *Cross-Entropy Loss*:

$$\mathcal{L}_{\text{CE}} = -\sum_{l \in \mathcal{D}^{\text{batch}}} \sum_{c=1}^{C} y_l^c \log \hat{y}_l^c, \quad (10)$$

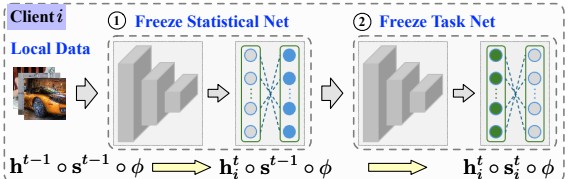

Figure 2: Two-stage training procedure for HStat-Net.

where $C$ is the number of classes, $y_l^c$ is the one-hot ground-truth label, and $\hat{y}_l^c$ is the predicted probability for class $c$. During each communication round $t$, client $i$ initializes local training from the global model $\mathbf{h}^{t-1} \circ \mathbf{s}^{t-1}$. As illustrated in Figure 2, training proceeds in two steps:

- **Step 1:** The Task Net is updated by minimizing $\mathcal{L}_{\text{CE}}$ with $\mathbf{s}^{t-1}$ fixed.
- **Step 2:** With $\mathbf{h}_i^t$ fixed, the Statistical Net is updated by minimizing $\mathcal{L}_{\text{Triplet}}$.

After local updates, each client transmits the composed transformation $\psi_i^t = \mathbf{h}_i^t \circ \mathbf{s}_i^t$ to the server, which performs uniform aggregation over the participating clients $\mathcal{N}^{(t)}$ to obtain the global $\psi^t$. This decoupled training strategy enables independent optimization of the Statistical Net and Task Net, mitigating objective interference and promoting stable convergence. It also facilitates efficient onboarding of new clients, as fine-tuning only the Task Net $\mathbf{h}$ is typically sufficient to achieve strong performance, further reducing computational and communication overhead [50].

**Experimental Validation.** We evaluate HStat-Net on the CIFAR-100 dataset [28] using 64 clients, where data is partitioned according to a Dirichlet distribution $\text{Dir}_{64}(\alpha)$ with $\alpha = 0.5$, representing moderate statistical heterogeneity [51, 52]. The CLIP model (ViT-B/32) [11] is used as a frozen feature extractor. We com-

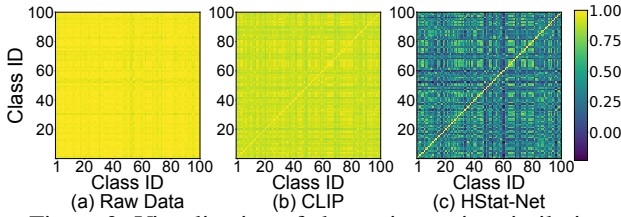

Figure 3: Visualization of class-wise cosine similarity.

pare three types of representations: raw inputs, CLIP features, and HStat-Net outputs. Figure 3 shows the class-wise cosine similarity across these representations. As features progress through the architecture, class separation becomes more pronounced. The average inter-class similarity decreases from 0.976 (raw inputs) to 0.908 (CLIP) and further to 0.339 (HStat-Net), indicating improved discriminability.

To quantitatively assess the quality of learned features, we compute Fisher's Criterion and Mutual Information (MI) [53, 54, 55] for the three representation types. As shown in Table 2, HStat-Net achieves the highest scores, yielding a 3.34× improvement in Fisher's Criterion and a

Table 2: Representation quality analysis.

| Method | Raw | CLIP | HStat-Net |
|--------|-----|------|-----------|
| Fisher | 0.149 | 0.480 | **1.602** |
| MI | 0.162 | 0.275 | **0.556** |

2.02× gain in MI compared to raw inputs. These results demonstrate that the hierarchical refinement introduced by HStat-Net produces a more structured and linearly separable feature space.

We further assess HStat-Net's generalization ability by simulating a deployment scenario in which ten previously unseen clients are introduced after initial training. These clients are not involved in the original training with $\alpha = 0.5$ and receive data drawn from $\text{Dir}_{10}(\alpha)$

Table 3: Performance on new clients under different data distributions.

| Method | $\alpha = 0.1$ | $\alpha = 0.5$ | $\alpha = 0.9$ |
|--------|--------------|--------------|--------------|
| Traditional | 23.78% | 12.90% | 9.67% |
| HStat-Net | **66.85%** | **55.64%** | **52.60%** |

with $\alpha \in \{0.1, 0.5, 0.9\}$, reflecting different levels of non-IID heterogeneity. Each client fine-tunes only its Task Net for one local epoch. As shown in Table 3, HStat-Net consistently outperforms the baseline $\mathbf{h} \circ \phi$ across all settings.

## 3.2 DevGuard: Deviance-based Guard Mechanism

While HStat-Net improves task utility by producing structured and linearly separable representations, the resulting stable feature space may be exploited by adversaries to insert subtle manipulations that retain local accuracy but deviate from global semantics. To mitigate this, we propose *DevGuard*, a server-side mechanism that detects semantic inconsistencies through statistical deviance analysis.

Unlike gradient-based defenses [20, 42, 56], DevGuard evaluates the alignment between client-level features and global class-wise representations.

**Representation-Driven Evaluation.** To capture semantic deviations introduced by malicious clients, each client $i$ computes class-wise feature centroids using its local Statistical Net $\mathbf{s}^{t-1}$ from the previous round and sends them to the server for aggregation:

$$\mathbf{r}_i^c = \frac{1}{n_i^c} \sum_{l=1}^{n_i^c} \mathbf{s}^{t-1}(\mathbf{z}_{i,l}^c), \tag{11}$$

where $\mathbf{z}_{i,l}^c$ is the $l$-th sample in class $c$, and $n_i^c$ is the number of local samples from that class. The server aggregates these vectors across clients using an element-wise median to obtain the global class-level representation:

$$\mathbf{r}_{\text{global}}^c = \text{median}\left(\{\mathbf{r}_i^c \mid i \in \mathcal{N}^{(t)}\}\right), \tag{12}$$

which provides a robust estimate of the shared semantic structure while suppressing outlier effects [19]. Given that HStat-Net produces linearly separable features, the Task Net $\mathbf{h}$ can be modeled as a generalized linear model (GLM) [57]. Specifically, the probability for class $c$ is computed as:

$$g(\mathbb{E}[Y]) = \mathbf{R}^\top \mathbf{w_h}, \tag{13}$$

where $\mathbf{R}$ is the input feature vector, $\mathbf{w_h}$ is the class-specific coefficient vector, and $g^{-1}$ is the inverse link function (e.g., softmax [58]). GLMs are widely used in tasks such as multinomial regression and anomaly detection [59, 60], offering both interpretability and robustness. These properties are especially valuable for evaluating client-level consistency in federated settings.

**Deviance-Based Client Evaluation.** Given the linearly separable representations generated by HStat-Net, DevGuard evaluates client reliability by applying deviance analysis under the GLM framework. Each client models the conditional distribution $\mathbb{P}(Y \mid \mathbf{R})$ using a linear transformation followed by a softmax activation:

$$\hat{y}_{\mathbf{r}}^c = \frac{\exp(\mathbf{r}^\top \mathbf{w_h}^c)}{\sum_{k=1}^C \exp(\mathbf{r}^\top \mathbf{w_h}^k)}, \tag{14}$$

where $\mathbf{r}$ is the input feature (e.g., $\mathbf{r}_{\text{global}}^c$), $\mathbf{w_h}^c$ is the coefficient vector for class $c$, and $C$ is the number of classes. Under this GLM formulation, we compute the deviance, a standard statistical metric that quantifies the goodness-of-fit by measuring the discrepancy between predicted probabilities and true labels [59]. For client $i$, the global representation $\mathbf{r}_{\text{global}}^c$ is passed through its Task Net to produce $\hat{y}_{\mathbf{r}}^c$, the predicted probability for class $c$. The true label is encoded as an indicator variable $y_{\mathbf{r}}^c$, which equals 1 if $c$ is the correct class and 0 otherwise. The log-likelihood for class $c$ is given by:

$$L_i^c = \sum_{c=1}^C y_{\mathbf{r}}^c \log(\hat{y}_{\mathbf{r}}^c), \tag{15}$$

where $\hat{y}_{\mathbf{r}}^c$ is the predicted probability produced by client's Task Net. In the saturated model, which represents a perfect fit to the data, the prediction for the correct class is $\hat{y}_{\mathbf{r}}^c = 1$, resulting in a log-likelihood of $L_{\text{saturated}}^c = \log(1) = 0$. The class-wise deviance residual is thus computed as:

$$\Delta_i^c = 2\left(L_{\text{saturated}}^c - L_i^c\right) = -2 \log\left(\hat{y}_{\mathbf{r}}^c\right). \tag{16}$$

To obtain a robust client-level reliability score, we aggregate the residuals across all classes using an entropy-inspired formulation:

$$\Delta_i = \sum_{c=1}^C \Delta_i^c \cdot \log\left(\Delta_i^c\right). \tag{17}$$

This approach amplifies the impact of large residuals, ensuring that clients with strongly misaligned predictions are penalized more heavily. A high $\Delta_i$ indicates that client $i$ deviates significantly from the global semantic structure, suggesting possible poisoning or manipulation.

**Adaptive Thresholding with Theoretical Guarantees.** Building on the deviance scores $\{\Delta_i\}_{i \in \mathcal{N}^{(t)}}$ computed for each client, we now describe how DevGuard distinguishes malicious participants using an adaptive thresholding strategy grounded in statistical analysis. Specifically, the scores are first sorted in ascending order:

$$\Delta_{[1]} \leq \Delta_{[2]} \leq \cdots \leq \Delta_{[n]}, \tag{18}$$

where $n = |\mathcal{N}^{(t)}|$ and $\Delta_{[i]}$ denotes the $i$-th smallest residual. Let $\mathcal{B}$ and $\mathcal{M}$ denote the sets of benign and malicious clients, respectively. We assume that the residuals from benign clients are centered around a lower mean $\mu_{\mathcal{B}}$, while those from malicious clients are centered around a higher mean $\mu_{\mathcal{M}}$, where $\mu_{\mathcal{M}} > \mu_{\mathcal{B}}$. This separation pattern can be expressed as:

$$\Delta_{[i]} = \begin{cases} \mu_{\mathcal{B}} + O_p(1), & \text{if } i \in \mathcal{B}, \\ \mu_{\mathcal{M}} + O_p(1), & \text{if } i \in \mathcal{M}, \end{cases} \tag{19}$$

where $O_p(1)$ denotes bounded variation due to random sampling or model variance. Our goal is to determine a threshold index $\hat{p}$ such that all clients with $\Delta_{[i]} > \Delta_{[\hat{p}]}$ are flagged as suspicious. To guide the threshold selection, we establish the following theoretical result:

**Theorem 1.** *Assume the deviance residuals of benign and malicious clients are drawn from distributions with means $\mu_{\mathcal{B}}$ and $\mu_{\mathcal{M}}$, respectively, and bounded variance $\sigma^2$, where $\mu_{\mathcal{M}} > \mu_{\mathcal{B}}$. Then, the total misclassification rate (TMR) is bounded by:*

$$TMR \leq \frac{4\sigma^2}{(\mu_{\mathcal{M}} - \mu_{\mathcal{B}})^2}. \tag{20}$$

This result provides a theoretical guarantee for separating benign and malicious clients based on their deviance scores. Since the exact means $\mu_{\mathcal{B}}$ and $\mu_{\mathcal{M}}$ are unknown, we approximate the optimal index $\hat{p}$ by evaluating all candidate indices $p \in \{1, 2, \ldots, n\}$ and selecting the one that minimizes the empirical upper bound in Equation (20).

To enhance robustness against noise and outliers, DevGuard employs a multi-step voting procedure. In each of $K$ steps, a random subset $\mathcal{N}_{\text{sub}}^{(t)} \subset \mathcal{N}^{(t)}$ is sampled, and corresponding global representations are recalculated. Within each step $k$, clients compute fresh residuals $\{\Delta_i^{(k)}\}$, and the threshold $\hat{p}_k$ is determined using the same optimization process. Clients with $\Delta_i^{(k)} > \Delta_{[\hat{p}_k]}$ are flagged as suspicious in that step. Final classification is based on majority voting:

$$\text{Votes}_i = \sum_{k=1}^{K} \mathbb{I}\left(\Delta_i^{(k)} > \Delta_{[\hat{p}_k]}\right) > \frac{K}{2},$$

where $\mathbb{I}(\cdot)$ is the indicator function. A client is classified as malicious if it is flagged in more than half of the steps (see Algorithm 1). This adaptive strategy combines statistical rigor with practical robustness, ensuring reliable detection under non-stationary or adversarial conditions.

## 3.3 Putting Everything Together

FEDRACE combines semantic representation learning on the client side with statistical evaluation on the server side to support reliable detection of malicious clients and robust model aggregation. The system is implemented using the FedScale platform [26] with PyTorch [61], and leverages GPU acceleration for efficient training and inference. In each communication round, after receiving class-wise centroids and corresponding model updates from participating clients, the server initiates a $K$-step detection procedure. In each step $k$, a random subset of clients $\mathcal{N}_{\text{sub}}^{(t)}$ is selected. The server aggregates centroids from this subset using element-wise median to form global class-level representations. It then evaluates each client by computing class-wise deviance residuals from its Task Net predictions, which are aggregated into a client-level score $\Delta_i$ to measure semantic consistency.

After calculating client scores, the server sorts the values and selects a threshold index $\hat{p}$ that minimizes the empirical bound on the TMR, as defined in Theorem 1. Clients with scores greater than the threshold, $\Delta_i > \Delta_{[\hat{p}]}$, are flagged as suspicious in that step. This procedure is repeated across $K$ steps. A client is ultimately classified as malicious if it is flagged in more than half of the steps. Only clients identified as benign are included in the global aggregation. The server-side computational complexity per round is $O\big(KN(C + \log N)\big)$, where $N$ is the number of clients and $C$ is the number of classes. This ensures that FEDRACE is scalable for large federated learning deployments.

Table 4: Performance of defense methods against different attacks, evaluated across multiple metrics.

| Dataset | Defense | Untargeted | | Targeted | | | | | |
|---|---|---|---|---|---|---|---|---|---|
| | | Min-Max | IPMA | TLFA | | ECBA | | DBA | |
| | | ACC | ACC | ASR | ACC | BA | ACC | BA | ACC |
| **CIFAR-100** | Multi-krum | $72.59_{0.27}$ | $76.16_{0.32}$ | $1.52_{0.10}$ | $75.93_{0.28}$ | $20.05_{0.11}$ | $76.03_{0.31}$ | $23.20_{0.28}$ | $75.68_{0.27}$ |
| | Trimmed-mean | $75.15_{0.35}$ | $76.43_{0.27}$ | $1.79_{0.25}$ | $75.83_{0.24}$ | $10.34_{0.26}$ | $76.53_{0.26}$ | $12.16_{0.29}$ | $76.65_{0.26}$ |
| | FLAIR | $73.07_{0.29}$ | $75.74_{0.27}$ | $0.61_{0.16}$ | $74.49_{0.30}$ | $1.30_{0.23}$ | $76.21_{0.32}$ | $0.96_{0.17}$ | $75.65_{0.28}$ |
| | FedRoLA | $76.05_{0.33}$ | $76.84_{0.28}$ | $11.92_{0.28}$ | $74.88_{0.29}$ | $39.28_{0.28}$ | $76.47_{0.30}$ | $2.89_{0.28}$ | $77.04_{0.27}$ |
| | FLShield | $76.86_{0.24}$ | $76.66_{0.25}$ | $2.27_{0.29}$ | $75.63_{0.28}$ | $1.67_{0.28}$ | $76.81_{0.27}$ | $1.46_{0.27}$ | $76.99_{0.31}$ |
| | FEDRACE | $\mathbf{76.69}_{0.32}$ | $\mathbf{76.99}_{0.32}$ | $\mathbf{0.07}_{0.10}$ | $\mathbf{77.02}_{0.33}$ | $\mathbf{0.06}_{0.11}$ | $\mathbf{76.98}_{0.31}$ | $\mathbf{0.36}_{0.23}$ | $\mathbf{77.21}_{0.31}$ |
| **Food-101** | Multi-krum | $52.31_{0.33}$ | $55.70_{0.27}$ | $2.07_{0.13}$ | $55.85_{0.27}$ | $20.22_{0.13}$ | $55.87_{0.28}$ | $49.13_{0.30}$ | $55.23_{0.29}$ |
| | Trimmed-mean | $54.37_{0.31}$ | $56.37_{0.31}$ | $2.34_{0.26}$ | $56.08_{0.28}$ | $27.58_{0.29}$ | $56.22_{0.32}$ | $30.84_{0.29}$ | $56.54_{0.29}$ |
| | FLAIR | $53.16_{0.30}$ | $54.27_{0.30}$ | $0.43_{0.15}$ | $52.09_{0.29}$ | $5.67_{0.30}$ | $55.24_{0.29}$ | $1.48_{0.25}$ | $53.33_{0.29}$ |
| | FedRoLA | $56.40_{0.29}$ | $55.59_{0.29}$ | $12.74_{0.29}$ | $54.10_{0.29}$ | $45.27_{0.26}$ | $56.16_{0.31}$ | $8.14_{0.28}$ | $56.51_{0.28}$ |
| | FLShield | $56.24_{0.29}$ | $56.07_{0.31}$ | $14.02_{0.32}$ | $54.76_{0.30}$ | $6.36_{0.29}$ | $56.25_{0.31}$ | $1.44_{0.28}$ | $56.65_{0.27}$ |
| | FEDRACE | $\mathbf{56.38}_{0.27}$ | $\mathbf{56.76}_{0.26}$ | $\mathbf{0.27}_{0.16}$ | $\mathbf{56.68}_{0.27}$ | $\mathbf{0.31}_{0.16}$ | $\mathbf{56.70}_{0.26}$ | $\mathbf{1.01}_{0.31}$ | $\mathbf{56.72}_{0.27}$ |
| **Tiny ImageNet** | Multi-krum | $71.04_{0.32}$ | $72.38_{0.28}$ | $0.63_{0.10}$ | $72.70_{0.27}$ | $19.27_{0.12}$ | $72.85_{0.27}$ | $45.71_{0.29}$ | $72.05_{0.28}$ |
| | Trimmed-mean | $71.95_{0.28}$ | $72.44_{0.29}$ | $0.95_{0.22}$ | $72.74_{0.28}$ | $33.06_{0.28}$ | $72.33_{0.30}$ | $35.09_{0.23}$ | $72.67_{0.25}$ |
| | FLAIR | $71.23_{0.35}$ | $72.59_{0.28}$ | $0.28_{0.19}$ | $70.58_{0.28}$ | $4.43_{0.28}$ | $71.89_{0.28}$ | $0.24_{0.15}$ | $70.91_{0.30}$ |
| | FedRoLA | $73.36_{0.21}$ | $72.78_{0.29}$ | $4.87_{0.27}$ | $71.92_{0.29}$ | $47.14_{0.28}$ | $72.73_{0.25}$ | $4.75_{0.28}$ | $73.13_{0.21}$ |
| | FLShield | $73.29_{0.24}$ | $73.19_{0.32}$ | $9.85_{0.28}$ | $71.84_{0.29}$ | $5.84_{0.28}$ | $73.11_{0.28}$ | $0.53_{0.19}$ | $73.21_{0.32}$ |
| | FEDRACE | $\mathbf{73.06}_{0.29}$ | $\mathbf{73.40}_{0.29}$ | $\mathbf{0.07}_{0.10}$ | $\mathbf{73.24}_{0.31}$ | $\mathbf{0.08}_{0.10}$ | $\mathbf{73.44}_{0.29}$ | $\mathbf{0.13}_{0.13}$ | $\mathbf{73.42}_{0.29}$ |

# 4 Experiments

## 4.1 Experimental Setup

**Datasets and Models.** We evaluate our framework on three widely used image classification benchmarks: CIFAR-100 [28], Food-101 [27], and Tiny ImageNet [25]. CIFAR-100 contains 100 classes with 600 images per class, divided into 500 training and 100 testing samples. Food-101 includes 101 food categories with 750 training and 250 testing images per class, and presents significant variation in appearance. Tiny ImageNet is a subset of ImageNet [62], consisting of 200 classes with 500 training and 50 validation images per class, offering a good trade-off between diversity and computational cost. For feature extraction, we use the CLIP model with a ViT-B/32 backbone and remove its final classification layer following standard transfer learning practice. The Statistical Net ($\mathbf{s}$) and the Task Net ($\mathbf{h}$) are implemented as single fully connected layers with output dimension $d = 256$, resulting in representations $\mathbf{r} \in \mathbb{R}^{256}$. To evaluate scalability, we also experiment with ResNet-152 [63] as an alternative backbone for $\phi$.

**Parameter Settings.** We simulate a FL system with $N = 64$ clients, selecting $n = 16$ clients randomly in each communication round. Following prior work [23, 29, 30], we assume $M = 16$ malicious clients, keeping the malicious ratio below 50%. These clients are randomly chosen in each round to reflect dynamic adversarial behavior. Client datasets are partitioned in a non-IID manner using a Dirichlet distribution with concentration parameter $\alpha = 0.5$. Local training uses a learning rate of 0.001, batch size 128, and three epochs per round across all datasets. All experiments are conducted on NVIDIA RTX A4500 GPUs and repeated with four random seeds (1, 12, 123, 1234).

**Baseline Attacks.** We evaluate FEDRACE under five representative attack scenarios in federated learning. The Min-Max Attack [23] generates updates that remain within acceptable bounds while introducing harmful behavior. The Inner Product Manipulation Attack (IPMA) [17] alters gradient directions to disrupt learning. The Targeted Label Flipping Attack (TLFA) [34] modifies labels to influence specific classification outcomes. The Edge-Case Backdoor Attack (ECBA) [36] embeds hidden triggers by leveraging rare or atypical input patterns. The Distributed Backdoor Attack (DBA) [18] involves coordination among multiple malicious clients to implant consistent backdoors.

**Baseline Defenses.** We compare our framework against five defense baselines: FLShield [21], FedRoLA [42], FLAIR [22], Trimmed-mean [19, 40], and Multi-Krum [20]. FLShield verifies local models using a reference dataset. FedRoLA analyzes update similarity to identify anomalies. FLAIR maintains dynamic reputations to weigh client contributions. Trimmed-mean discards extreme values for robust aggregation. Multi-Krum selects a subset of updates with minimal pairwise distances to exclude outliers. We omit the Median aggregator since recent studies [42, 56] report its performance is generally comparable to Trimmed-mean.

Table 5: Comparison of detection performance (TPR/FPR) under various scenarios.

| Dataset | Defense | Untargeted | | Targeted | | |
|---|---|---|---|---|---|---|
| | | Min-Max | IPMA | TLFA | ECBA | DBA |
| **CIFAR-100** | Multi-krum | 0.783/0.223 | 0.817/0.192 | 0.853/0.148 | 0.804/0.197 | 0.732/0.263 |
| | FLAIR | 0.847/0.352 | 0.882/0.318 | 0.943/0.357 | 0.912/0.383 | 0.867/0.342 |
| | FedRoLA | 0.931/0.118 | 0.907/0.133 | 0.648/0.353 | 0.613/0.387 | 0.583/0.412 |
| | FLShield | 0.935/0.183 | 0.927/0.157 | 0.912/0.173 | 0.887/0.187 | 0.872/0.213 |
| | FEDRACE | 0.987/0.072 | 0.983/0.018 | 0.973/0.012 | 0.977/0.032 | 0.968/0.101 |
| **Food-101** | Multi-krum | 0.773/0.227 | 0.808/0.204 | 0.842/0.163 | 0.793/0.218 | 0.724/0.267 |
| | FLAIR | 0.834/0.367 | 0.857/0.343 | 0.927/0.354 | 0.893/0.392 | 0.847/0.364 |
| | FedRoLA | 0.934/0.127 | 0.893/0.147 | 0.634/0.373 | 0.578/0.423 | 0.547/0.442 |
| | FLShield | 0.922/0.193 | 0.918/0.168 | 0.904/0.182 | 0.873/0.214 | 0.863/0.227 |
| | FEDRACE | 0.992/0.063 | 0.987/0.023 | 0.988/0.018 | 0.983/0.024 | 0.978/0.112 |
| **Tiny ImageNet** | Multi-krum | 0.787/0.214 | 0.827/0.184 | 0.857/0.143 | 0.813/0.187 | 0.743/0.248 |
| | FLAIR | 0.864/0.338 | 0.887/0.313 | 0.947/0.342 | 0.923/0.368 | 0.883/0.334 |
| | FedRoLA | 0.948/0.113 | 0.917/0.123 | 0.663/0.338 | 0.617/0.384 | 0.587/0.403 |
| | FLShield | 0.953/0.174 | 0.932/0.153 | 0.918/0.164 | 0.893/0.183 | 0.878/0.204 |
| | FEDRACE | 0.993/0.048 | 0.988/0.013 | 0.987/0.028 | 0.992/0.023 | 0.983/0.085 |

## 4.2 Experimental Results

**Main Results.** Table 4 presents the performance of all defense methods under both untargeted and targeted attacks across CIFAR-100, Food-101, and Tiny ImageNet. For untargeted attacks, such as Min-Max and IPMA, traditional aggregation methods like Multi-Krum and Trimmed-mean achieve moderate accuracy (e.g., 72.59% and 75.15% on CIFAR-100 under Min-Max), while more recent defenses such as FLShield and FedRoLA offer improved results. FEDRACE consistently achieves the highest accuracy under IPMA, with 76.99% on CIFAR-100, 56.76% on Food-101, and 73.40% on Tiny ImageNet. For targeted attacks, metrics such as ASR and BA reveal clearer differences. While FedRoLA yields an ASR of 11.92% on CIFAR-100 under TLFA, and FLShield reduces this to 2.27%, both still allow some attack success. FLAIR lowers ASR further (e.g., 0.61%) but at the cost of high FPR. In contrast, FEDRACE reduces ASR and BA to below 0.4% across most datasets and attacks, while maintaining competitive accuracy (e.g., 77.21% under DBA on CIFAR-100).

Table 5 reports the detection performance of all methods in terms of true positive rate (TPR) and false positive rate (FPR). FEDRACE achieves the best balance, with TPRs consistently above 0.97 and FPRs below 0.1 in most scenarios. For instance, under TLFA on CIFAR-100, FEDRACE achieves a TPR of 0.973 with a FPR of only 0.012, significantly outperforming FedRoLA (TPR: 0.648, FPR: 0.353) and FLAIR (TPR: 0.943, FPR: 0.357). Similar trends hold across Food-101 and Tiny ImageNet. These results demonstrate that FEDRACE not only enhances robustness against diverse attacks but also offers reliable detection with minimal false alarms. Beyond standard attack scenarios, additional tests against adaptive and stealthy attacks, including IBA [64], A3FL [65], and our Statistical-Net-only variant, show that FEDRACE keeps attack success below 2.5% and true-positive rates above 93%, demonstrating robustness even under adaptive adversaries.

**Evaluation of Detection Threshold.** Theorem 1 guides the selection of the detection threshold. Table 6 shows the error $|\hat{p} - p^*|$ in threshold estimation under different attack types and datasets. For targeted attacks like ECBA, the error is very small (at most 0.033). For untargeted attacks such as Min-Max and for DBA, the error is higher but still acceptable. These results show that FEDRACE can estimate the threshold accurately in most cases.

Table 6: Evaluation of threshold estimation.

| $|\hat{p} - p^*|$ | CIFAR-100 | Food-101 | ImageNet |
|---|---|---|---|
| **Min-Max** | $0.113_{0.26}$ | $0.102_{0.26}$ | $0.168_{0.24}$ |
| **IPMA** | $0.029_{0.12}$ | $0.043_{0.14}$ | $0.032_{0.16}$ |
| **TLFA** | $0.041_{0.20}$ | $0.038_{0.18}$ | $0.039_{0.13}$ |
| **ECBA** | $0.024_{0.12}$ | $0.033_{0.14}$ | $0.022_{0.11}$ |
| **DBA** | $0.148_{0.28}$ | $0.064_{0.16}$ | $0.192_{0.25}$ |

**Evaluation of Scalability.** We evaluate the scalability of FEDRACE by replacing the CLIP feature extractor with ResNet-152. As shown in Figure 4, FEDRACE maintains strong detection performance across all datasets. For untargeted attacks, the TPR remains above 0.98, while the FPR stays below 0.07. For targeted attacks such as TLFA and ECBA, TPR is consistently between 0.97 and 0.99, with FPR between 0.02 and 0.03. Even under the more challenging DBA attack, TPR stays above 0.97, with only a minor increase

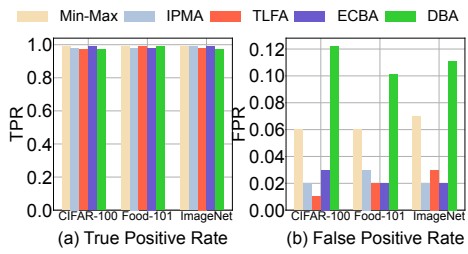

Figure 4: FEDRACE under ResNet-152.

in FPR. These results confirm that HStat-Net ensures robust detection across different backbone models, making FEDRACE independent of specific feature extractors.

**Evaluation of Parameters.** Table 7 presents the impact of two key parameters in FEDRACE: the client subset size $|\mathcal{N}_{\text{sub}}^{(t)}|$ and the number of voting steps $K$. The results show that smaller subset sizes, such as $n/4$, generally lead to lower detection performance. For example, under the DBA attack with $K = n/2$, the TPR is $0.887$ when $|\mathcal{N}_{\text{sub}}^{(t)}| = n/4$, while it improves to $0.968$ when the subset size increases to $n/2$. Further increasing the subset size to $3n/4$ does not consistently yield better performance and sometimes results in higher FPR. Across different attacks, the best trade-off is achieved when both $|\mathcal{N}_{\text{sub}}^{(t)}|$ and $K$ are set to $n/2$, which provides a high TPR and a low FPR. These findings indicate that a moderate subset size and a balanced number of detection steps are sufficient to ensure reliable performance without introducing unnecessary overhead.

Table 7: Detection performance of FEDRACE under different parameter settings.

| $\mid\mathcal{N}_{\text{sub}}^{(t)}\mid$ | $K$ | Min-Max | | TLFA | | DBA | |
|---|---|---|---|---|---|---|---|
| | | TPR | FPR | TPR | FPR | TPR | FPR |
| | $n/4$ | 0.883 | 0.167 | 0.862 | 0.178 | 0.843 | 0.192 |
| | $n/3$ | 0.902 | 0.152 | 0.887 | 0.156 | 0.868 | 0.173 |
| $n/4$ | $n/2$ | 0.923 | 0.128 | 0.901 | 0.143 | 0.887 | 0.165 |
| | $2n/3$ | 0.938 | 0.113 | 0.917 | 0.128 | 0.902 | 0.149 |
| | $3n/4$ | 0.942 | 0.108 | 0.922 | 0.122 | 0.908 | 0.142 |
| | $n/4$ | 0.943 | 0.103 | 0.921 | 0.087 | 0.903 | 0.142 |
| | $n/3$ | 0.968 | 0.084 | 0.952 | 0.043 | 0.941 | 0.118 |
| $n/2$ | $n/2$ | 0.987 | 0.072 | 0.973 | 0.012 | 0.968 | 0.101 |
| | $2n/3$ | 0.992 | 0.066 | 0.981 | 0.008 | 0.975 | 0.088 |
| | $3n/4$ | 0.994 | 0.063 | 0.984 | 0.007 | 0.978 | 0.082 |
| | $n/4$ | 0.932 | 0.142 | 0.913 | 0.092 | 0.892 | 0.157 |
| | $n/3$ | 0.958 | 0.112 | 0.937 | 0.063 | 0.923 | 0.138 |
| $3n/4$ | $n/2$ | 0.975 | 0.096 | 0.952 | 0.045 | 0.943 | 0.127 |
| | $2n/3$ | 0.981 | 0.092 | 0.958 | 0.042 | 0.948 | 0.122 |
| | $3n/4$ | 0.983 | 0.088 | 0.961 | 0.038 | 0.951 | 0.118 |

**Impact of Data Heterogeneity.** We evaluate the effect of data heterogeneity by varying the Dirichlet concentration parameter $\alpha$ on Tiny ImageNet. As shown in Figure 5, we consider three settings: extreme non-IID ($\alpha = 0.1$), moderate non-IID ($\alpha = 0.5$), and near-IID ($\alpha = 0.9$). Across all settings and attack types, FEDRACE consistently achieves a true positive rate of at least $0.968$, indicating strong detection performance even under severe distribution shifts. Moreover, the false positive rate decreases as $\alpha$ increases, suggesting that more balanced data distributions improve detection precision by reducing semantic divergence across clients.

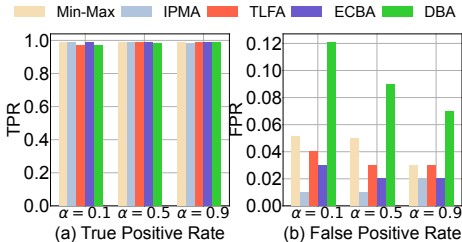

Figure 5: Detection performance under different non-IID settings.

**Robustness to Attack Scale.** To assess robustness under varying attack intensity, we vary the number of malicious clients $M$ on Tiny ImageNet, testing with $M = 8$ (12.5%), $M = 16$ (25%), and $M = 24$ (37.5%) out of 64 total clients. As shown in Figure 6, FEDRACE maintains a TPR above $0.97$ across all scenarios and attack types. Although the FPR slightly increases with a larger number of attackers, the overall performance remains stable. These results confirm that FEDRACE is resilient to changes in both data heterogeneity and adversarial scale, making it effective in diverse federated learning environments.

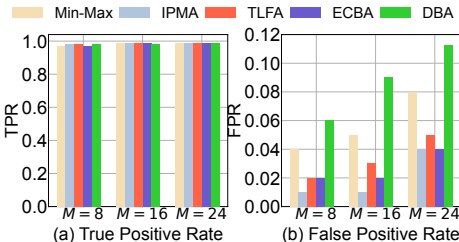

Figure 6: Detection performance under different numbers of malicious clients.

## 5   Conclusions and Limitations

In conclusion, FEDRACE is a robust FL framework that integrates HStat-Net for representation learning with GLM-based deviance analysis for secure client evaluation. It leverages pre-trained models to improve generalization and identifies malicious clients through statistical residuals. While effective for classification tasks, extending this framework to domains such as text generation or retrieval requires new strategies for semantic representation. Future work will focus on adapting the model to support broader tasks and enabling robust, multi-modal FL in diverse settings.

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

# A Appendix

## A.1 Algorithm Overview of the Multi-step Voting

---

**Algorithm 1** Multi-step Voting

---

**Require:** Client set $\mathcal{N}^{(t)}$, voting steps $K$, subset size $\beta$
**Ensure:** Detection results indicating malicious clients
1: Initialize Votes $\leftarrow$ zeros($|\mathcal{N}^{(t)}|$), with $|\mathcal{N}_{\text{sub}}^{(t)}| = |\mathcal{N}^{(t)}|/2$
2: **for** $k = 1$ to $K$ **do**
3:      $\mathcal{N}_{\text{sub}}^{(t)} \leftarrow$ RandomSelect($\mathcal{N}^{(t)}, |\mathcal{N}_{\text{sub}^{(t)}}|$)
4:      **for** each class $c$ **do**
5:          $\mathbf{r}_{\text{global}}^c \leftarrow$ median$\left( \left\{ \mathbf{r}_i^c \mid i \in \mathcal{N}_{\text{sub}}^{(t)} \right\} \right)$
6:      **end for**
7:      **for** each client $i \in \mathcal{N}^{(t)}$ **do**
8:          **for** each class $c$ **do**
9:              $\Delta_i^c \leftarrow -2 \log\left( \hat{y}_{\mathbf{r}}^c \right)$
10:        **end for**
11:        $\Delta_i \leftarrow \sum_{c=1}^{C} \Delta_i^c \cdot \log\left( \Delta_i^c \right)$
12:      **end for**
13:      Sort residuals: $\Delta_{[1]} \leq \cdots \leq \Delta_{[n]}$
14:      For each candidate threshold $p \in \{1, \ldots, n\}$, compute estimates: $\mu_{\mathcal{B},p}, \mu_{\mathcal{M},p}$, and $\sigma_p^2$
15:      Select adaptive threshold $\hat{p}_k \leftarrow \arg\min_p \frac{4\sigma_p^2}{(\mu_{\mathcal{M},p} - \mu_{\mathcal{B},p})^2}$
16:      **for** each client $i \in \mathcal{N}^{(t)}$ **do**
17:          **if** $\Delta_i > \Delta_{[\hat{p}_k]}$ **then**
18:              Votes$[i] \leftarrow$ Votes$[i] + 1$
19:          **end if**
20:      **end for**
21: **end for**
22: Declare clients as malicious: $\hat{\mathcal{M}} = \left\{ i \mid \text{Votes}[i] > \frac{K}{2} \right\}$
23: Perform global aggregation using benign clients:
$$\psi^t = \frac{1}{|\mathcal{N}^{(t)} \setminus \hat{\mathcal{M}}|} \sum_{i \in \mathcal{N}^{(t)} \setminus \hat{\mathcal{M}}} \psi_i^t.$$

---

## A.2 Details for Equation 16

In this work, we treat the task net $\mathbf{h}$ in HStat-Net as a Generalized Linear Model, where feature representations refined by the statistical net $\mathbf{s}$ are linearly transformed and passed through a softmax function to produce class probabilities. The linearly separable representation space generated by statistical net $\mathbf{s}$ renders the task net equivalent to a multinomial logistic regression. Following previous works [57, 66, 67], we analyze model fit using deviance residuals, which quantify the model's alignment with expected prediction.

For each aggregated global representation $\mathbf{r}_{\text{global}}^c$ of class $c$, the true class label is denoted by the indicator variable $y_{\mathbf{r}}^c$:

$$y_{\mathbf{r}}^c = \begin{cases} 1, & \text{if global representation belongs to class } c; \\ 0, & \text{otherwise.} \end{cases} \tag{21}$$

The task net trained on client $i$ predicts the probability that representation $\mathbf{r}_{\text{global}}^c$ belongs to class $c$, denoted by $\hat{y}_{\mathbf{r}}^c$. The log-likelihood function of the trained task net for representation $\mathbf{r}_{\text{global}}^c$ is:

$$L_i^c = \sum_{c=1}^{C} y_{\mathbf{r}}^c \log(\hat{y}_{\mathbf{r}}^c) = \log(\hat{y}_{\mathbf{r}}^c), \tag{22}$$

which resembles the log-likelihood of multinomial logistic regression in the GLM framework. Assume a saturated model that perfectly fits the representation, where the predicted probabilities match the observed labels, which are one-hot encoded:

$$\hat{y}_{\mathbf{r},\text{saturated}}^c = y_{\mathbf{r}}^c, \tag{23}$$

Then, the log-likelihood for the saturated model is:

$$L_{\text{saturated}}^c = \sum_{c=1}^{C} y_{\mathbf{r}}^c \log(\hat{y}_{\mathbf{r},\text{saturated}}^c) = \log(\hat{y}_{\mathbf{r},\text{saturated}}^c), \tag{24}$$

Since $y_{\mathbf{r}}^c$ is either 0 or 1, and $\log(1) = 0$, only the terms where $y_{\mathbf{r}}^c = 1$ contribute, leading to a log-likelihood of zero for the saturated model:

$$L_{\text{saturated}}^c = 0. \tag{25}$$

The deviance residual $\Delta_i^c$ is defined as twice the difference between the log-likelihoods of the saturated model and the task net:

$$\Delta_i^c = 2(L_{\text{saturated}}^c - L_i^c) = -2L_i^c. \tag{26}$$

The deviance residual for each representation on a task net is given by:

$$\Delta_i^c = -2\log(\hat{y}_{\mathbf{r}}^c) \tag{27}$$

This shows that the deviance residual for each observation depends solely on the predicted probability of the true class. A larger deviance residual indicates a worse fit for that representation, potentially signaling malicious activity on client $i$'s task net.

### A.3 Proof of Theorem 1

*Proof.* Our detection algorithm sorts clients based on their deviance residuals $\Delta_i$. We aim to determine a threshold $\mathcal{T}$ such that clients with $\Delta_i \leq \mathcal{T}$ are classified as benign, and clients with $\Delta_i > \mathcal{T}$ are classified as malicious. To analyze the misclassification rates, we define the following:

(i) The false positive rate (FPR) is the probability that a benign client is misclassified as malicious:

$$\text{FPR} = \mathbb{P}(\Delta_i > \mathcal{T} \mid i \in \mathcal{B}).$$

(ii) The false negative rate (FNR) is the probability that a malicious client is misclassified as benign:

$$\text{FNR} = \mathbb{P}(\Delta_i \leq \mathcal{T} \mid i \in \mathcal{M}).$$

Our goal is to choose the threshold $\mathcal{T}$ that minimizes the total misclassification rate (TMR), defined as

$$\text{TMR} = \pi_{\mathcal{B}} \cdot \text{FPR} + \pi_{\mathcal{M}} \cdot \text{FNR},$$

where $\pi_{\mathcal{B}} = 1 - \pi_{\mathcal{M}}$ is the proportion of benign clients.

Since we do not assume a specific distribution for the deviation residuals, we apply Chebyshev's inequality to bound the probabilities of misclassification. For any random variable $X$ with expected value $\mu$ and variance $\sigma^2$, Chebyshev's inequality states that for any $\delta > 0$,

$$\mathbb{P}(|X - \mu| \geq \delta) \leq \frac{\sigma^2}{\delta^2}.$$

Applying Chebyshev's inequality to the deviation residuals:

For benign clients:

$$\mathbb{P}(|\Delta_i - \mu_{\mathcal{B}}| \geq \delta \mid i \in \mathcal{B}) \leq \frac{\sigma^2}{\delta^2}.$$

For malicious clients:

$$\mathbb{P}(|\mu_{\mathcal{M}} - \Delta_i| \geq \delta \mid i \in \mathcal{M}) \leq \frac{\sigma^2}{\delta^2}.$$

We choose the threshold $\mathcal{T}$ as the midpoint between the expected residuals of benign and malicious clients:

$$\mathcal{T} = \frac{\mu_{\mathcal{B}} + \mu_{\mathcal{M}}}{2}.$$

This choice sets $\delta = \frac{\mu_{\mathcal{M}} - \mu_{\mathcal{B}}}{2}$. Substituting $\delta$ into the bounds:

$$\text{FPR} \leq \frac{4\sigma^2}{(\mu_{\mathcal{M}} - \mu_{\mathcal{B}})^2},$$

$$\text{FNR} \leq \frac{4\sigma^2}{(\mu_{\mathcal{M}} - \mu_{\mathcal{B}})^2}.$$

Therefore, the total misclassification rate is bounded by

$$\text{TMR} = \pi_{\mathcal{B}} \cdot \text{FPR} + \pi_{\mathcal{M}} \cdot \text{FNR} \leq \frac{4\sigma^2}{(\mu_{\mathcal{M}} - \mu_{\mathcal{B}})^2}.$$

This bound shows that the total misclassification rate decreases as the square of the difference between the mean residuals increases. Specifically, as the separation $\mu_{\mathcal{M}} - \mu_{\mathcal{B}}$ becomes larger relative to the variance $\sigma^2$, the misclassification rate approaches zero. $\qquad\square$

