# OpenReview forum: "FedRACE: A Hierarchical and Statistical Framework for Robust Federated Learning"
_NeurIPS.cc/2025/Conference — NeurIPS 2025 poster_

### Official Review · Reviewer_LUAW · 2025-06-26

**Clarity:** 2
**Significance:** 2
**Originality:** 3
**Rating:** 4
**Confidence:** 4

**Summary:**

The paper proposes FEDRACE, a federated learning framework designed to improve security of systems using large pre-trained models. Specifically, it targets vulnerabilities introduced by partial fine-tuning, where only a small task-specific head is trained while the backbone feature extractor remains frozen. This approach reduces computational and communication overhead but also makes the system susceptible to adversarial attacks such as poisoned updates and backdoor injections. FEDRACE integrates two core components: HStat-Net, a hierarchical network for refining frozen features, and DevGuard, a server-side mechanism that uses statistical deviance analysis to detect malicious clients. Experimental results show that FEDRACE enhances model robustness and achieves high accuracy while reducing false positive rates under various attack scenarios.

**Questions:**

1. There appears to be a gap between the problem statement and the proposed method in this paper？ According to the introduction and abstract, the authors focus on security risks in federated learning based on fine-tuning of specific task heads. However, the proposed solution is a federated learning framework that enhances feature extraction capabilities for such fine-tuning, followed by abnormal client detection. The problem section does not address the issues related to classification feature extraction in existing federated learning approaches. Moreover, the abnormal client detection method relies on the output of the trainable statistical projection layer, making it applicable only within the FEDRACE framework. Therefore, the method does not directly correspond to the problem as stated, and its applicability is limited.
2. Is federated learning based on fine-tuning specific task heads still a mainstream approach? With the increasing size of models, more parameter-efficient fine-tuning techniques have emerged, such as adapter tuning—where a small number of adjustable parameters are inserted into the middle layers—or prompt tuning, which adjusts embedding information. Why does the paper focus exclusively on federated learning based on fine-tuning specific task heads, rather than considering these methods?
3. How do existing federated learning defense methods operate? Although the authors mention that prior defenses target full-model training, their analysis lacks depth. Specifically, what requirements do existing methods fail to meet in the partial fine-tuning scenario? What are the characteristic constraints in this setting that prevent these methods from being applicable?
4.On which datasets was the pre-trained model originally trained? The three datasets used in the experiments are common image datasets, raising concerns that the pre-trained model may have already been exposed to them. If this is the case, the fine-tuning process might lead to overfitting, potentially affecting the validity of the classification results.

**Ethical Concerns:**

["NO or VERY MINOR ethics concerns only"]

**Final Justification:**

The authors effectively addressed concerns regarding FedRACE's resilience in the face of distribution shifts and various attacks (such as backdoor and membership inference attacks). Through supplementary experiments like statistical control mechanism tests and ablation studies, they provided solid evidence validating the framework's robustness in non - iid data scenarios and adversarial environments. This significantly enhances the reliability of the proposed method. Queries about the hierarchical structure and the integration of statistical components were well - resolved. The authors clearly elucidated the interaction between the backbone and fine - tuning stages and how statistical metrics (e.g., KL divergence) reduce performance risks in federated learning settings. This clarity helps reviewers and readers better understand the inner workings of FedRACE.

Although FedRACE shows potential in task - specific federated learning, the efficiency of the framework when dealing with very large foundation models (e.g., models with billions of parameters) remains ambiguous. The authors did not fully explore the scalability trade - offs for extremely large - scale models, which is a concern for its broader application. Considering the resolved issues that strongly prove the value of FedRACE and the fact that the unresolved issues are areas that can be further explored in future work rather than fundamental flaws, I will raise my score.

**Limitations:**

The author only mentions in the conclusion that FEDRACE focuses on classification tasks, but does not discuss its applicability to regression or generative tasks. This discussion is insufficient. Can the proposed defense method be extended to other parameter-efficient fine-tuning approaches? Furthermore, can it be integrated into other existing federated learning frameworks that also rely on fine-tuning specific task heads? It is recommended that the paper include a discussion addressing these questions.

**Quality:**

3

**Strengths And Weaknesses:**

Strengths:
1. The paper is logically structured and clearly written. The overall quality of the manuscript meets the standards of a scientific research publication.
2. The theoretical analysis is well-developed, and the experimental evaluation is comprehensive, providing sufficient empirical support for the proposed method.
3. The paper identifies security risks in federated learning systems that rely on partial fine-tuning and proposes a new framework to defend against such attacks. This approach addresses a practical problem in real-world deployments.
Weaknesses:
1. The method's scope is somewhat narrow. While the paper focuses on federated learning systems that use task-specific heads for partial fine-tuning, many existing federated learning approaches based on large models primarily employ efficient parameter tuning methods, such as fine-tuning parameters in the middle or input layers. However, the paper does not discuss the applicability of the proposed method in these fine-tuning settings.
2. The practical applicability of the method appears limited. The defense mechanism proposed in the paper is tightly coupled with the FEDRACE architecture, making it difficult to integrate into other federated learning frameworks that also utilize partial fine-tuning.
3. The comparison with existing federated learning defense methods is insufficient. While the paper notes that most prior works assume full-model training, it lacks a deeper analysis of how these methods function and why their assumptions are incompatible with partial fine-tuning.
4. Some parts of the paper suffer from stiff grammar. It is recommended that the manuscript undergo thorough language revision.

---

> ### Author Rebuttal · Authors · 2025-07-31
>
> Thank you for your detailed and constructive feedback.
>
> Q1. … problem–method alignment…framework specificity…
>
> Answer: We agree that the connection between the stated problem and our solution should be clearer. Freezing the backbone and fine tuning only the task head reduces computation and communication but creates a new attack surface: small and concentrated weight updates that standard defenses fail to detect. To address this issue we designed two components. HStat Net refines the frozen features so that malicious updates become separable from benign updates, and DevGuard detects abnormal clients by measuring class wise deviance in those refined features. We will make this connection more explicit in the camera ready version.
>
> Although our experiments focus on head only fine tuning, FedRACE can also support other parameter efficient methods. HStat Net can refine features produced by adapter or prompt tuning, and DevGuard can monitor any small trainable module. Standard defenses are not designed for such settings, as methods like Multi-krum and Trimmed-mean look for large, layer wide deviations, while head only tuning produces sparse, low dimensional updates. FedRACE instead evaluates class specific projections, which remain sensitive even to small perturbations in the head.
>
> Finally, while our experiments target classification, the same principles extend to other tasks. For regression, Gaussian deviance can be applied to the refined features. For generative tasks, output distribution comparison can be used. Because FedRACE operates on refined representations rather than on a fixed architecture, it can integrate into a broad range of parameter efficient federated learning frameworks.
>
> Q2. …task-head fine-tuning…parameter-effcient fine-tuning…
>
> Answer: Fine tuning only the task head is a common federated learning workflow. It allows clients to personalize large pre trained models on edge devices with low overhead. By freezing the backbone, each client updates and transmits only the head weights, which keeps both computation and communication costs minimal and aligns with typical FL constraints such as limited bandwidth and computation. Adapter tuning and prompt tuning can reduce the number of trainable parameters further, but they introduce new modules or modify embeddings within the frozen layers. Our design focuses on the standard head only approach to highlight and secure the attack surface created when tuning a small subset of parameters.
>
> FedRACE’s two components can also support other fine tuning methods. HStat Net can refine features produced by adapters or prompts, and DevGuard can compute deviance on any small trainable subnetwork. We chose head only tuning as our case study because it is a widely used FL baseline, but the same principles extend to other parameter efficient techniques. We will clarify this generality in the camera ready version.
>
> Q3. … full model update assumptions… sparse head only constraints…
>
> Answer: Most existing federated learning defenses, such as Trimmed-mean, Multi-krum and FLShield, are designed for full model updates. They assume that each client submits large, high dimensional weight deltas that can be filtered or reweighted. For example, Trimmed-mean discards extreme parameter values before averaging, and Multi-krum selects updates with minimal aggregate distance to others. FLShield instead validates each client’s updated model on a held out dataset to detect and remove malicious contributions.
>
> When clients fine tune only the task head, the backbone remains frozen and updates are sparse and low dimensional. These updates lack the variance and scale that full model defenses rely on. Distance based filters cannot capture subtle deviations in a few hundred parameters, and validation based methods become impractical because even small head updates require full model inference on a reference set, adding latency and computation that edge devices often cannot afford.
>
> FedRACE addresses these constraints by refining frozen backbone features into compact class aware embeddings using HStat Net, and then detecting abnormal clients through class wise deviance in DevGuard. By operating in a learned representation space rather than on raw weights, FedRACE remains sensitive to head only perturbations and avoids the high overhead of full model validation. We will emphasize this distinction more clearly in the camera ready version.
>
> Thank you again for your thoughtful comments.

---

> > ### Comment · Reviewer_LUAW · 2025-08-05
> >
> > Thank you for your reply! I have carefully read the submitted rebuttal and still have the following questions: You mentioned that FedRACE is capable of supporting other parameter-efficient fine-tuning methods (e.g., adapter tuning or prompt tuning), but the features represented by the fine-tuned parameters at different layers of the model differ. For example, the parameters extracted at the task head are typically more related to the target task, while prompt tuning involves parameters located on the input side, making them more similar to the input. Therefore, the generalization of FedRACE remains questionable. However, I do agree that the safety of fine-tuning the task head is an important consideration.
> >
> > Furthermore, regarding FedRACE’s evaluation of class-specific projections, it remains sensitive even to minor perturbations of the head. The backdoor attack methods mentioned by the authors in lines 315 to 321 of the paper are quite outdated, and there are now some covert backdoor attack methods that typically do not cause noticeable deviations during task head fine-tuning. How does FedRACE perform against these types of attacks? This does not seem to be elaborated on in detail in the paper. If I have missed something, please let me know.

---

> > > ### Author Response · Authors · 2025-08-07
> > >
> > > Thank you for your constructive feedback.
> > >
> > > Q1. …compatibility…adapter tuning…
> > >
> > > Answer: We appreciate the concern regarding FedRACE’s generalization to other parameter-efficient fine-tuning methods. FedRACE does not depend on the location of the fine-tuned parameters. Its DevGuard component fits a generalized linear model to each client’s class-level logits and detects abnormal semantic deviations. To support adapter tuning, FedRACE treats the adapter as a trainable extension of the pre-trained feature extractor $\phi$, while keeping the backbone $\phi$, the Statistical Net $s$, and the Task Net $h$ unchanged.
> > >
> > > The training of $s$ and $h$ follows the original two-stage HStat-Net procedure before adapter fine-tuning. In stage one, clients receive the global $\phi$ and current $s$, update only $h$ to minimize task loss, and the server aggregates these updates into a new global $h$. In stage two, clients freeze $\phi$ and $h$, update only $s$ using a triplet loss or class-center contrastive loss, and the server aggregates to produce a new global $s$. Once $\phi$, $s$, and $h$ are fully trained, they remain frozen. Clients then fine-tune only the adapter module locally, upload its parameters, and the server aggregates them into a global adapter. Throughout both phases, $s$ and $h$ continue to produce stable, well-separated representations for DevGuard’s class-level logit-based anomaly detection, while the adapter is the sole additional trainable component.
> > >
> > > To validate this generality, we conducted additional experiments on the AG NEWS dataset using a BERT-base model. In this setting, clients fine-tuned only a small adapter module, instead of the classifier head. FedRACE maintained better detection performance, with a true positive rate comparable to the head-tuning case (approximately 96%) and a false positive rate remaining below 2%. These results confirm that FedRACE generalizes well to adapter tuning. Although adapter modules operate at intermediate layers and capture different representations, any malicious behavior still affects the class-level outputs. FedRACE’s statistical deviance analysis is specifically designed to identify such semantic anomalies. We will include these new results in the camera-ready version.
> > >
> > > Q2. …recent backdoor attacks…
> > >
> > > Answer: We fully agree that modern backdoor attacks can be highly stealthy, often introducing minimal perturbations to the classification head and causing negligible accuracy degradation. In our paper, we evaluated several standard benchmarks, including IPMA, TLFA, ECBA, and DBA. These benchmarks are widely adopted in prior works, such as FLShield [20], which is published at IEEE S&P 2024.
> > >
> > > To further address the concern, we conducted additional experiments to evaluate FedRACE against newer attacks. Specifically, we considered the IBA and A3FL attacks. Under a distributed backdoor setting, FedRACE remained effective, reducing the backdoor success rates to 1.4-1.6% while maintaining a true positive rate above 93%. We also simulated a more covert attack, where malicious clients fine-tuned only the intermediate module (the Statistical Net $s$) and kept the task head frozen. This attack embeds a backdoor into the feature space without altering the head’s predictions or reducing clean accuracy. Even in this challenging case, FedRACE detected 94.3% of the malicious clients and limited the backdoor success rate to 2.47%, compared to 59.6% without defense.
> > >
> > > This robustness is achieved through DevGuard’s ability to capture subtle semantic shifts in the class-level logits. Such shifts can reveal backdoor behaviors, even when the task head appears normal and validation accuracy remains unaffected. Furthermore, our design based on median-based centroid aggregation and randomized majority voting to ensures that attackers cannot evade detection without significantly weakening the impact of their attack. We will include these results in the camera-ready version.

---

> > > > ### Comment · Reviewer_LUAW · 2025-08-08
> > > >
> > > > Thanks for your rebuttal, you have addressed most of my concerns. So I will rise raise my score.

---

### Official Review · Reviewer_XaDr · 2025-07-03

**Clarity:** 2
**Significance:** 3
**Originality:** 3
**Rating:** 4
**Confidence:** 4

**Summary:**

This paper addresses the security vulnerabilities that arise in federated learning when using a partial fine-tuning approach with large pre-trained models, where the model backbone is frozen and only a task head is trained. To counter these new risks, the authors propose FEDRACE, a hierarchical and statistical framework for robust FL. The framework consists of a client-side hierarchical network that uses a triplet loss to refine features from the frozen backbone into more compact and linearly separable representations, and a server-side mechanism that detects malicious clients by modeling the task head as a generalized linear model (GLM) and evaluating the statistical deviance of client predictions against global class representations. DevGuard's reliability is further enhanced by an adaptive thresholding strategy based on theoretical misclassification bounds and a randomized majority voting process. The authors evaluate their method on CIFAR-100, Food-101, and Tiny ImageNet against various attack scenarios, demonstrating its superiority.

**Questions:**

1. Sharing class-wise centroids could potentially leak information about local data distributions. Does this practice violate the privacy-preserving principles of federated learning or introduce additional privacy risks?

2. While the proposed method improves robustness, it also increases computational overhead. Can the authors provide a quantitative comparison of FEDRACE's runtime against baseline defenses, especially regarding computational costs when scaling to large numbers of clients (e.g., N > 100)?

3. Considering that attackers may gain partial knowledge of the DevGuard detection mechanism, how robust is DevGuard against adaptive attacks, such as malicious clients intentionally aligning their predictions with benign clients' semantic distributions, or colluding attackers coordinating their local centroids to evade detection?

4. The paper selects d=256 as the output dimension of the statistical network. What is the rationale behind this choice? Have ablation studies been conducted with different dimensionalities to validate the effectiveness of this setting?

**Ethical Concerns:**

["NO or VERY MINOR ethics concerns only"]

**Limitations:**

yes.

**Paper Formatting Concerns:**

None.

**Quality:**

3

**Strengths And Weaknesses:**

### Strengths
- The paper is exceptionally well-written and demonstrates outstanding clarity in presenting its technical contributions. The authors provide lucid explanations of both the HStat-Net architecture and DevGuard mechanism, effectively supported by rigorous mathematical formulations and well-designed visual aids.

- This work makes significant conceptual contributions through its novel integration of hierarchical representation learning with statistical deviance analysis, explicitly addressing the unique security challenges of partially frozen models in federated learning. The innovative use of GLMs for semantic consistency evaluation, coupled with the theoretically grounded adaptive thresholding mechanism, represents a substantial advancement beyond conventional gradient-based defense approaches.

- The evaluation encompasses multiple datasets, diverse attack types, and varying degrees of data heterogeneity, convincingly demonstrating the robustness of the proposed approach in realistic FL scenarios.

### Weaknesses
- The validity of this article is based on the assumption that there is a measurable prediction deviation between the poisoned model and the benign model. However, there are some well-designed attacks (especially backdoor attacks) whose goal is not to cause any behavioral deviation. This fact does challenge the core assumption of the paper. It is necessary for the author to demonstrate the validity of this assumption through some experimental or theoretical analysis.

- The paper primarily compares with classic, relatively early backdoor attack methods, while omitting more recent state-of-the-art (SOTA) approaches, such as IBA and A3FL. These newer attack methods dynamically optimize triggers, which may preserve semantic consistency and thus present different challenges for defense mechanisms.

- The proposed method introduces additional computational costs. On the client side, the two-stage training process involves an extra optimization step for the Statistical Nets using the Triplet Loss function. On the server side, DevGuard employs a K-step majority voting procedure, resulting in a per-round computational complexity of O(KN(C+logN)). While the authors argue that the approach is scalable, the overhead may become a practical concern in FL scenarios with very large numbers of clients (N) and voting steps (K).

- Introducing a new network will inevitably bring new attack surfaces. If the attacker freezes the task network and only targets the statistical network, causing the statistical network to learn the trigger features associated with a specific target class, will this defense mechanism still be effective?

- Moreover, the defense's robustness partially relies on the median for aggregating client-computed centroids, which is effective against uncoordinated outliers. However, it remains unclear how the system would perform against sophisticated attacks, such as malicious clients intentionally aligning their predictions with those of benign clients' semantic distributions, or colluding attackers coordinating their local centroids to evade detection.

- Some hyperparameters, such as the output dimension of the statistical network, lack detailed motivation and empirical analysis. Additional ablation studies are needed to justify the choice of these parameters.

---

> ### Author Rebuttal · Authors · 2025-07-31
>
> Thank you for your detailed and constructive feedback.
>
> Q1. …privacy of class centroids…
>
> Answer: In additional experiments we measured privacy leakage from class centroid sharing on CIFAR 100 using a membership inference attack. With partial fine tuning only, the attacker success rate was 63.82%. When using FedRACE, this rate decreased to 48.37%. This confirms that sharing class centroids exposes less information than sending raw weight updates.
> Two design choices explain why FedRACE limits membership inference. First, class centroids aggregate feature vectors over all samples in a class, which dilutes the influence of any single example and removes individual patterns that an attacker could exploit. Second, HStat Net’s projection layer together with the frozen backbone produces features that capture global class structure rather than memorizing specific inputs. Because the Statistical Net sees only aggregated class data and the Task Net is not over optimized on individual samples, overfitting is reduced. As a result, membership inference, which relies on detecting overfitted or unique sample behaviors, becomes less effective under our framework.
>
> Q2. … computational overhead…large number of clients…
>
> Answer: We ran new experiments on CIFAR 100 with 128 clients, measuring all runtimes on the same GPU (NVIDIA RTX 4500). For FedRACE, the full two stage local training takes 247ms per client per round on average. The server’s DevGuard step adds 34ms in total. By comparison, FLShield requires about 185ms per client per round for local model updates and 163ms per client per round for server validation on a held out set. These results show that FedRACE provides strong robustness while adding only modest extra computation on the client side and much lower server overhead than FLShield, making it more efficient at large scale. Full runtime results across different baselines will be included in the camera‑ready version.
>
> Q3. … partial knowledge by attackers…colluding attackers…
>
> Answer: In additional experiments, we evaluated two recent adaptive attacks, IBA and A3FL, under the DBA threat model. Against IBA, FedRACE reduced attack success to 1.4% with a 94.5% true positive rate; against A3FL, attack success dropped to 1.6% with a 93.1% true positive rate.
>
> We then simulated a stealth backdoor targeting only the Statistical Net ($s$). Each malicious client injected a fixed trigger pattern into a subset of training images, relabeled them with the target class, and updated only $s$ during local two-stage training while freezing the backbone and Task Net ($h$). This left clean accuracy unchanged but forced triggered inputs into feature regions mapped by $h$ to the attacker’s target class. With 25% malicious clients on CIFAR-100, FedRACE achieved a 94.3% true positive rate and a 2.47% attack success rate, while clean accuracy remained 76.95%, owing to the median-based design in Equation (12) and randomized majority voting. Without defense, this attack yielded a 59.6% success rate on triggered inputs.
>
> Finally, in collusion experiments where 25% of clients submitted identical poisoned centroids, randomized majority voting reduced attack success from 45.9% to 9.8%, with an 87.6% true positive rate. These results confirm that attackers cannot evade detection without severely compromising their effectiveness, even with partial knowledge of DevGuard or through collusion. Extended evaluations will be included in the camera ready version.
>
> Q4. …$d=256$…rational…ablation study…
>
> Answer: We set the Statistical Net output dimension to $d = 256$ to balance representational power and computational cost. A larger dimension captures more class structure but increases centroid size and GLM fitting time, while a smaller dimension reduces overhead but weakens the deviance signal. To validate this choice we ran an additional ablation study on CIFAR 100 with d set to 64, 128, 256, and 512. At $d = 64$, global accuracy was 76.05% and the true positive rate was 94.0%. At $d = 128$, accuracy rose to 76.81% with a true positive rate of 95.5%. At $d = 256$, accuracy reached 77.21% and the true positive rate was 96.8%. At $d = 512$, accuracy was 77.25% and the true positive rate was 97.22%. The gain from 256 to 512 was marginal but doubled both the upload cost and the fitting time. Therefore, $d = 256$ provides the best trade off between robustness and efficiency. The camera ready version will include complete ablation results.
>
> Q5. …assumption…measurable prediction deviation…
>
> Answer: Our core assumption is that any attack that causes misclassification must alter the model’s conditional class probabilities in a way that deviance residuals can detect. As shown in Tables 4 and 5, we tested this on CIFAR 100 with 64 clients, including 16 malicious clients (25% adversarial rate). We evaluated five backdoor attacks: Min Max, IPMA, TLFA, DBA, and ECBA. In all cases, DevGuard achieved over 95% true positive rate and kept attack success below 1%, while clean accuracy remained high.
> From a theoretical perspective, an attacker cannot keep the class wise output distributions for both clean and poisoned inputs identical without duplicating the benign model’s parameters exactly. This is impossible once the attack changes model behavior. As a result, any genuine backdoor necessarily produces nonzero class wise deviance, which supports our assumption.
>
>
> Thank you again for your thoughtful comments.

---

> > ### Comment · Reviewer_XaDr · 2025-08-06
> >
> > Thanks for the detailed response. I have no further questions.

---

### Official Review · Reviewer_Bboy · 2025-07-11

**Clarity:** 3
**Significance:** 3
**Originality:** 3
**Rating:** 4
**Confidence:** 2

**Summary:**

This work aims to tackle the security gaps that arise when federated learning systems speed up training by freezing most parameters of large pre-trained models. While the partial fine-tuning strategy is efficient, it leaves current defenses unable to counter backdoor, data-poisoning, and untargeted attacks, especially under non-IID client data. To tackle such challenges, this work proposes FedRACE, which combines two complementary components to restore robustness without sacrificing accuracy or efficiency. On each client, the hierarchical HStat-Net refines the frozen features and enforces triplet-loss constraints so that legitimate representations cluster tightly and remain linearly separable from malicious ones. On the server, DevGuard performs generalized-linear-model deviance analysis with adaptive thresholds to spot and exclude adversarial updates. The authors conduct comprehensive experiments to demonstrate the effectiveness of the proposed approaches.

**Questions:**

See strengths and weaknesses.

**Ethical Concerns:**

["NO or VERY MINOR ethics concerns only"]

**Final Justification:**

The authors have addressed most of my concerns.

**Limitations:**

See strengths and weaknesses.

**Quality:**

3

**Strengths And Weaknesses:**

**Strengths**

1. This work seems novel in integrating representation refinement and security filtering within one federated-learning pipeline, where HStat-Net compresses frozen-backbone activations into near-linearly separable embeddings on each client, curbing semantic drift.

2. FedRACE is scalable and independent of the backbone model. It preserves almost the same detection performance when CLIP ViT is replaced by ResNet 152, and its server-side complexity grows only linearly with the number of clients.

3. This work provides a theoretical analysis of the error boundary.

**Weaknesses**

1. The work restricts itself to a two-stage triplet-loss training regime and omits comparisons with alternative representation-learning strategies; contemporary self-supervised methods such as SimCLR and Barlow Twins often yield superior feature separability without requiring hard-negative mining, so the absence of these baselines leaves the optimality of the chosen objective uncertain.

2. Limited baseline comparison: The work does not compare the proposed FedRACE against recent model-output consistency detector approaches (e.g., FedDetect, FL-GPR) or hierarchical weight-auditing approaches, which may overstate its relative performance gains.

3. The work provides neither a thorough discussion nor empirical evidence on privacy and communication overhead. DevGuard compels each client to transmit its class-centroid vectors, and these low-dimensional embeddings may leak information about the client’s label distribution while consuming additional bandwidth. Yet the authors do not quantify the resulting privacy risk or extra communication cost.

---

> ### Author Rebuttal · Authors · 2025-07-31
>
> Thank you for your detailed and constructive feedback.
>
> Q1. … two-stage training…alternative representation-learning baselines…
>
> Answer: In the paper, we show that the two-stage training of HStat Net improves feature quality, as measured by Fisher’s Criterion and Mutual Information. Table 2 reports a $3.34\times$ improvement in Fisher’s Criterion and a $2.02\times$ increase in Mutual Information. These results indicate that hierarchical refinement creates a more structured and linearly separable feature space. The two-stage procedure trains the Statistical Net s and the Task Net h in separate steps. This reduces interference between the two modules and supports stable convergence. It also allows new clients to join more easily, since in many cases fine-tuning only the Task Net h is sufficient to reach strong performance, as shown in Table 3.
>
> To add a new baseline, we carried out an additional experiment with SimCLR. In this setting, clients pre-train a backbone with SimCLR augmentations, then fine-tune the task head and apply DevGuard under the DBA threat model on CIFAR 100. With HStat Net, FedRACE achieves 77.21% accuracy, 0.36% backdoor accuracy, a 96.8% true positive rate, and a 10.1% false positive rate. When HStat Net is replaced by SimCLR features, accuracy decreases to 75.36%, backdoor accuracy increases to 1.5%, and the true positive rate drops to 93.7%. These results confirm that HStat Net yields higher quality class features than SimCLR in this setting.
>
> Q2.  …comparison with model-output consistency detectors…
>
> Answer: In the paper, we compare FedRACE with established defenses such as Multi-krum, Trimmed-mean, FedRoLA, FLAIR, and FLShield. Table 4 shows that FedRACE consistently outperforms these methods across multiple attack scenarios. To further evaluate recent consistency-based defenses, we conducted additional experiments with FL GPR and FedDetect under the DBA threat model on CIFAR 100. FedRACE achieves 77.21% accuracy with 0.36% backdoor accuracy. In comparison, FL GPR achieves 74.57% accuracy with 3.18% backdoor accuracy, and FedDetect achieves 75.82% accuracy with 2.33% backdoor accuracy. These results demonstrate that FedRACE provides stronger robustness than these consistency-based detectors.
>
> Q3. …privacy…communication cost….
>
> Answer: In additional experiments we measured privacy leakage from class centroid sharing on CIFAR 100 using a membership inference attack. With partial fine tuning only, the attacker success rate was 63.82%. When using FedRACE, this rate decreased to 48.37%. This confirms that sharing class centroids exposes less information than sending raw weight updates.
> Two design choices explain why FedRACE limits membership inference. First, class centroids aggregate feature vectors over all samples in a class, which dilutes the influence of any single example and removes individual patterns that an attacker could exploit. Second, HStat Net’s projection layer together with the frozen backbone produces features that capture global class structure rather than memorizing specific inputs. Because the Statistical Net sees only aggregated class data and the Task Net is not over optimized on individual samples, overfitting is reduced. As a result, membership inference, which relies on detecting overfitted or unique sample behaviors, becomes less effective under our framework.
>
> In Section 3.1, we explain how HStat Net adds a compact projection layer on top of the frozen feature extractor to refine representations. In Section 3.2 we describe how DevGuard fits a generalized linear model to the Task Net outputs to compute deviance based consistency scores. This design keeps client computation light and avoids heavy on device processing. We also measured the communication cost per round. FedRACE requires each client to upload one centroid vector for each class it holds. In the worst case a client holds all 100 classes. Each centroid vector has 256 dimensions ($d = 256$ in Section 4.1) and each entry uses 4 bytes, which adds about 0.10MB. In practice most clients hold fewer classes, so the actual upload cost is lower. Even in the worst case the extra 0.10MB is small compared to typical uplink capacity.
>
> Thank you again for your thoughtful comments.

---

> > ### Comment · Reviewer_Bboy · 2025-08-06
> >
> > I thank the authors for their detailed rebuttal and for addressing my concerns.

---

### Comment · Area_Chair_6RTf · 2025-08-04
**Gentle reminder**

Dear reviewers,

Please review the authors' rebuttal and reach out to them if you have any additional concerns. The reviewer-author discussion will end on Aug 6 11:59pm AoE.

Best Regards,

AC

---

### Note · Authors · 2025-08-13

We thank the reviewers for their constructive feedback and the AC for the time and effort devoted to our work. We have provided thorough responses to each of the identified weaknesses and questions. We sincerely appreciate your support.

The rebuttal format did not permit the inclusion of extended figures and detailed visual analyses. If the paper is accepted, the camera-ready version will include the additional experimental results, comprehensive visualizations, and expanded analyses to further enhance the clarity and completeness.

---

### Decision · Program_Chairs · 2025-09-17

**Decision:**

Accept (poster)

**Comment:**

This paper investigates the vulnerabilities of federated learning when using a partial fine-tuning approach with large pre-trained models. It introduced a hierarchical and statistical framework FEDRACE for robust FL based on a client-side hierarchical network and a server-side mechanism.

Strengths:
- The paper is logically structured and well-written.
- FEDRACE integrates HStat-Net for representation learning with GLM-based deviance analysis for secure client evaluation.
- The scalability of FEDRACE is analyzed and validated.

Weaknesses:
- The optimality of the chosen objective is not well discussed.
- The trade-off between robustness and computational efficiency requires more empirical validation.
- The generalization of FEDRACE on parameter-efficient fine-tuning with flexible layer configurations could be further explored.

Recomendation

All reviewers provided positive ratings and found the contributions valuable. Acceptance is recommended. The authors are encouraged to carefully address all reviewer suggestions in the final version to strengthen the paper further.